# Correlation between DNA Methylation and Cell Proliferation Identifies New Candidate Predictive Markers in Meningioma

**DOI:** 10.3390/cancers14246227

**Published:** 2022-12-17

**Authors:** Sébastien Hergalant, Chloé Saurel, Marion Divoux, Fabien Rech, Celso Pouget, Catherine Godfraind, Pierre Rouyer, Stéphanie Lacomme, Shyue-Fang Battaglia-Hsu, Guillaume Gauchotte

**Affiliations:** 1UMR Inserm 1256 NGERE (Nutrition, Génétique et Exposition aux Risques Environnementaux), Université de Lorraine, 54000 Nancy, France; 2Department of Biopathology CHRU-ICL, CHRU, 54500 Nancy, France; 3Department of Neurosurgery, CHRU, 54500 Nancy, France; 4CRAN, CNRS, Université de Lorraine, 54000 Nancy, France; 5Neuropathology, CHU of Clermont-Ferrand, UMR INSERM/Université d’Auvergne U1071, 63000 Clermont-Ferrand, France; 6Centre de Ressources Biologiques, BB-0033-00035, CHRU, 54500 Nancy, France

**Keywords:** genome-wide DNA methylation, meningioma, methylome, proliferation signature, biomarkers, survival, Ki-67, MCM6

## Abstract

**Simple Summary:**

In adults, meningioma is the most common primary tumor of the brain. It is classified into three clinical grades of aggressiveness. Whereas disease recurrence after surgery and survival are associated with grade, it is worth investigating proliferation at a molecular level to identify markers capable of improving the clinical management of meningioma. In this study, we explore the DNA methylation profiles of 48 tumors of various grades and conduct statistical analyses on several proliferation indices and markers, such as mitotic index, grade, and Ki-67 or MCM6 expression levels. We identify differential methylation profiles between grades, loci highly correlated with cell growth and division, and a specific methylation signature of regulatory regions persistently associated with proliferation indices, grade, and survival. Finally, we report candidate genes under the control of these regions with potential prognostic and therapeutic value and deserving clinical evaluation.

**Abstract:**

Meningiomas are the most common primary tumors of the central nervous system. Based on the 2021 WHO classification, they are classified into three grades reflecting recurrence risk and aggressiveness. However, the WHO’s histopathological criteria defining these grades are somewhat subjective. Together with reliable immunohistochemical proliferation indices, other molecular markers such as those studied with genome-wide epigenetics promise to revamp the current prognostic classification. In this study, 48 meningiomas of various grades were randomly included and explored for DNA methylation with the Infinium MethylationEPIC microarray over 850k CpG sites. We conducted differential and correlative analyses on grade and several proliferation indices and markers, such as mitotic index and Ki-67 or MCM6 immunohistochemistry. We also set up Cox proportional hazard models for extensive associations between CpG methylation and survival. We identified loci highly correlated with cell growth and a targeted methylation signature of regulatory regions persistently associated with proliferation, grade, and survival. Candidate genes under the control of these regions include *SMC4*, *ESRRG*, *PAX6*, *DOK7*, *VAV2*, *OTX1*, and *PCDHA*-*PCDHB*-*PCDHG*, i.e., the protocadherin gene clusters. This study highlights the crucial role played by epigenetic mechanisms in shaping dysregulated cellular proliferation and provides potential biomarkers bearing prognostic and therapeutic value for the clinical management of meningioma.

## 1. Introduction

Meningiomas are the most common primary tumors of the central nervous system in adults. The annual incidence rate ranges from 1.3/100,000 to 7.8/100,000 for cerebral meningiomas, with a tendency towards constant augmentation over the past few years [1]. The widely adopted WHO (World Health Organization) classification divides meningiomas into 15 subtypes and 3 grades of malignancy, mainly based upon histology [2]. Grade 1, 2, and 3 meningiomas represent about 70%, 20–30%, and 1–3% of reported cases, respectively. They correlate with recurrence risk (7–25% for grade 1, 29–59% for grade 2, and 50–94% for grade 3 [3]), as well as with 5-year and 10-year overall survival [4,5]. However, the histological criteria are rather subjective and are often associated with significant interobserver bias [6,7]; more reliable markers can thus improve the adequacy of treatments based on tumor grade. The last WHO classification (5th Edition, 2021) integrated a *TERT* promoter mutation or a homozygous deletion of *CDKN2A* and/or *CDKN2B* as new criteria for the recognition of grade 3 meningiomas. The standard of care is as follows: when deemed adequate, most patients undergo surgery, whereas adjuvant therapy is not systematic; for grade 2 and grade 3 meningioma, after surgery, conformational radiotherapy is recommended, particularly in cases of incomplete resection in grade 2 and in all grade 3 cases [8]. To date, no drug therapy has been validated for meningioma treatment. 

The discovery of new molecular targets may present new therapeutic options in meningioma management. The study of the molecular landscape in meningioma is thus an important issue. However, until recently, few genetic variations have been described. These included the earliest finding of chromosome 22q deletion, which causes the loss of the tumor suppressor gene *NF2* [9], the inactivation of which was observed in about half of the meningiomas studied [10]. More recently, several genes with recurrent mutations were identified in meningioma, including proapoptotic E3 ubiquitin ligase TNF receptor-associated factor 7 (*TRAF7*), pluripotency transcription factor Kruppel-like factor 4 (*KLF4*), proto-oncogene v-Akt murine thymoma viral oncogene homolog 1 (*AKT1*), Hedgehog pathway-signaling member “smoothened” (*SMO*), and phosphatidylinositol-4,5-bisphosphate 3-kinase catalytic subunit A (*PIK3CA*). Approximately 40% of sporadic meningiomas harbor at least one of these variations [11,12]. Inhibitors of *SMO*, *AKT1*, and *PIK3CA* are therefore of therapeutic interest [8,10]. Other recently identified mutations are linked to a phosphatase tensin homolog on chromosome 10 (*PTEN*), as well as cyclin kinases *CDKN2A*/*CDKN2B,* main tumor suppressor genes predominantly implicated in meningioma progression [13,14,15]. However, approximately 20% of meningiomas present no known oncogenic mutation [10]. Other genetic rearrangements may be implicated; these include copy-number alteration and chromosomal abnormalities, both of which are associated with higher grade and poor tumor prognosis [16,17,18,19]. These unbalanced profiles can impact genes involved in cell cycle maintenance and progression, dysregulate major functional pathways, activate oncogenes, and inactivate tumor suppressor genes [13,20]. 

Recently, DNA methylation (DNAm) profiles have been studied to elaborate new prognostic classifications [21,22,23,24]. Two were landmark studies that greatly advanced our understanding of the role of DNAm in meningioma. One study investigated meningioma genome-wide DNAm patterns and classified them into three distinct and clinically relevant methylation classes (benign, intermediate, and malignant) and six methylation subclasses (benign-1, benign-2, benign-3, intermediate-A, intermediate-B, and malignant). This approach more efficiently predicted tumor recurrence than the WHO classification [23]. The other study identified four key molecular/phenotypic features associated with meningioma malignancy based on an integrative analysis of multi-omic data gathered from DNAm, somatic point mutations, copy-number aberrations, and mRNA abundance. These features presented immunogenic, benign *NF2* wild-type, hypermetabolic, and proliferative characteristics of the tumor tissue and could be used to determine the most appropriate therapeutic strategy. This study further associated independent immunohistochemical markers with each molecular group. For instance, a high expression of MCM proteins—from the helicase complex involved in DNA replication, which can be used as a proliferation marker—was discovered to be characteristic of the proliferative group, with levels correlating with poor prognosis. This finding is consistent with our own observation that a high MCM6 index correlates with shorter progression-free survival [25], underscoring the crucial role of cell cycle progression and proliferation in meningioma progression. 

Taken together, these insights prompted us to investigate the correlation between DNAm and meningioma cell proliferation and evaluate their levels of dependency. To achieve this, we studied tumor cell proliferation based on the immunohistochemical markers Ki-67 and MCM6, which are associated with histopathological factors such as WHO grade and mitotic index on one hand and genome-wide DNAm profiles of a meningioma cohort composed of 48 tumor tissues of various grades on the other hand. Our goal was to identify specific genes correlating with/presenting differentially methylated regions as a function of tumor proliferation. To conclude the potential clinical application of our results, we evaluated the association between DNAm and survival in an attempt to facilitate the discovery of potential new prognostic markers and/or therapeutic targets.

## 2. Materials and Methods

### 2.1. Population and Clinical Data

Forty-eight samples from surgical meningioma resections analyzed in the Department of Pathology of Nancy University Hospital (CHRU Nancy, France) between 2006 and 2018, with available frozen tissue, were randomly included, focusing on high-grade meningioma associated with a grade 1 control group. All samples were anonymized. The study was conducted in accordance with local ethical guidelines. 

For each sample, we collected the WHO grade and other histopathological data from the pathology reports and clinical data from the medical records. Main clinical variables of interest were age, gender, localization of the tumor, quality of the excision (complete or not), treatment by chemotherapy or/and radiotherapy, recurrence of the meningioma and vital status. Because the date of first symptoms or diagnostic imaging was not available for all the patients, the diagnostic date was considered as the date of surgery. The progression date was considered as the day of the radiological exam during which progression was noted. The last consultation date and the last news date were also collected from the medical records. They were used to calculate overall survival (OS) and progression-free survival (PFS).

### 2.2. Histopathology

The WHO grade was established by a neuropathologist based on the WHO 2021 classification. The mitotic count per 10 high power fields (HPF; 1.6 mm^2^) was assessed independently by two pathologists, and the mean value was calculated. The level of agreement between the two raters was measured with the intraclass correlation coefficient. Both analyses were performed on hematoxylin, eosin, and saffron (HES) slides, blinded to the clinical and molecular data.

### 2.3. Immunohistochemistry

For each case, all immunohistochemical (IHC) staining was performed from the same block of formalin-fixed, paraffin-embedded (FFPE) tissue, which was selected after reviewing all the HES slides. Slides were manually prepared by paraffin sections of 4 µm, followed by deparaffinization, rehydration, and antigen retrieval. IHC staining was performed using the following antibodies: MCM6 (mouse monoclonal antibody, clone H-8, sc-3936-16, 1/2000 dilution, Santa Cruz) and Ki-67 (mouse monoclonal antibody, clone MIB-1, GA62661, prediluted, Dako, Agilent). Immunochemistry was performed with a Dako Omnis (Dako Agilent) automate using an Envision Flex revelation system (Dako Agilent). The labeling index (LI) for MCM6 and Ki-67 was calculated as the percentage of tumor cells with nuclear staining counted among a total of 1000 tumoral cells. 

### 2.4. DNA Methylation Analysis

DNA was extracted with a Macherey Nagel DNA extraction kit (Macherey-466 Nagel, Düren, Germany). After qualitative control, 900 ng of the extracted DNA was used to perform analyses with an Illumina MethylationEPIC BeadChip following the manufacturer’s instructions, as previously described [24,25]. Raw data files (IDAT) were generated and used for downstream bioinformatics with the minfi package in R v4.1. The EPIC microarray interrogates 850K CpG sites, enabling a genome-wide methylome study including proximal promoters, distal regulatory regions, gene bodies, and intergenic features. Numerous preparation and filtering steps for quality control (as described in [26]), including technical checks and CpG removal from X and Y chromosomes, led to a 787,087 CpG working dataset. Normalization was carried out following the FunNorm procedure [27].

Supervised analyses included statistical modeling with empirical Bayes for case/control studies (R limma package) and reported the differentially methylated CpGs, with *p*-values for each comparison adjusted for false discovery rate (FDR) following the Benjamini–Hochberg procedure. Differentially methylated regions (DMRs) between groups were determined using the same linear models with the dmrcate function of the R package DMRcate (parameters: lambda = 1000, C = 3, less than a 1000 bp gap, and at least 3 CpGs). Unbiased functional annotations on ontological terms (gene ontology, GO) were achieved at CpG and DMR levels with the R package missMethyl [28]. Spearman’s correlation tests were used for correlation analyses between CpG methylation levels and quantitative variables (mitotic index: number of mitoses, Ki-67 and MCM6 labeling indices: proliferative marker expressions). For each 787087 CpG, correlative metrics (Spearman’s Rho, *p*-value, and FDR by Benjamini–Hochberg) and beta-value statistics (average, median, standard deviation, methylation range max/min, and Q3/Q1) were computed and reported, along with CpG island (CGI) and gene contexts. By default, beta value is expressed from 0 (demethylated) to 1 (methylated), whereas methylation differentials, changes, and ranges are expressed in % and represent the real proportion of beta-value change: a 20% change signifies a 0.2 increase/decrease in beta value. 

Cox proportional hazard models were used to evaluate the association between CpG methylation levels and survival, either for PFS or OS analyses (R survival and survminer packages). After selection of the top CpGs with univariate setups in PFS in OS, multivariate analyses were conducted using age and grade as covariates. For each 787,087 CpG, associative metrics (Wald test *p*-values and effect size) were computed and reported, along with CGI and gene annotations. The proportional hazard assumption was tested on each fitted univariate model. Two statistics were considered: the *p*-value of non-random distribution for the methylation variable (CpG) and the global *p*-value for the model. When both *p*-values > 0.05, the CpG was retained for further multivariate evaluation. Proportional hazard assumptions were not rechecked for age and sex, as we only considered the statistical significance of the methylation covariate.

In every statistical approach, an FDR < 0.05 was considered significant. Supervised analyses (statistical modeling and DMR search) were conducted on methylation M values. Unsupervised analyses (hierarchical clustering) and correlative studies were performed with methylation B values (beta values).

Samples also underwent brain tumor classification and copy-number variation (CNV) estimation according to Capper et al. [29] and Sahm et al. [23]. Raw methylation files (IDAT) were uploaded to the MolecularNeuropathology.org server using the v11b4 classifier for brain tumor classification and MNGv2.4 for meningioma subtype classification. The evaluations of copy-number aberrations (along with CNV plots), *CDKN2A/B* loss, and *PTEN* loss were extracted from the CNV profile section of the generated report. 

## 3. Results

### 3.1. Clinicopathological Data 

Our cohort was composed of 48 patients with tumor samples of diverse WHO 2021 grades, including grade 1 (21%; 10/48), grade 2 (atypical; 69%; 33/48), and grade 3 (anaplastic; 10%; 5/48) meningiomas. In comparison with the original pathology reports, the grade changed in one case due to the detection of a homozygous *CDKN2A/B* deletion, upgrading the case from grade 2 to 3. The median age was 57 years, with a male-to-female ratio of 0.5. Meningioma tissues were mainly localized in the convexity (69%) and skull base (25%). No neoadjuvant treatment was delivered. Twenty-two patients (46%) underwent radiotherapy (postoperative and/or after progression; Appendix A), on patient (2%) received post-progression chemotherapy, and four patients received antiangiogenic (bevacizumab) therapy. Disease progression occurred in 19 cases (40%; median PFS time: 39 months (16;55)). Nine patients died during the follow-up, with a median OS time of 52 months (31;95). The mean mitotic index was 5.4 mitoses/mm^2^ (grade 1: 0.5/1.6 mm^2^; grade 2: 4.9/1.6 mm^2^; grade 3: 23/1.6 mm^2^), with good inter-observer agreement (intraclass correlation coefficient: 0.75). Average Ki-67 and MCM6 labeling indices were 21% and 51%, respectively (Table 1). 

### 3.2. Molecular Data Based on Molecular Neuropathology Classifiers and Copy-Number Variations

On MolecularNeuropathology.org, the brain tumor classification was successfully run on all 48 samples, reporting the methylation class “meningioma” with a valid score (calibrated score ˃ 0.9) for all samples (100%). The additional algorithm for meningioma subtype identification classified 31 meningiomas (65%) with a calibrated score ˃ 0.9: 13 in the benign class, 14 in the intermediate class, and 4 in the malignant class (Table 2). Grade 1 meningiomas were classified in benign and intermediate classes (80% and 20%, respectively). Among the 19 grade 2 cases with a calibrated score > 0.9, 5 were classified in benign (15%), 12 in intermediate (36%), and 2 in malignant (6%) classes. 

Copy-number variations (CNVs) identified by DNA methylation (DNAm) in the cohort of 48 meningiomas are represented in Figure 1. The total number of CNVs (complete or partial gain/deletion) averaged at 7.2 ± 5.7. The CNV profiles showed a homozygous loss of *CDKN2A*/B, loss of *PTEN*, and/or *NF2* genes in 0%, 0%, and 50% of grade 1 meningiomas; 0%, 42%, and 91% of grade 2 meningiomas; and 40%,100%, and 100% of grade 3 meningiomas, respectively (Table 2).

### 3.3. DNA Methylation and WHO Grade

At CpG resolution, differential analyses yielded 2099 significant hits between high-grade (grades 2 and 3; *n* = 38) and low-grade (grade 1; *n* = 10) meningiomas (Figure 2), with 1158 hypomethylated and 941 hypermethylated CpGs (moderated t-test; FDR < 0.05) (Appendix A). Of the total 2099 CpGs with FDR < 0.05, 2064 (98.3%) presented with a beta-value change > 1%, 1752 (83.5%) had a methylation differential > 5%, and 1641 (78.2%) showed > 10% change. Overall, hypomethylation occurred at already unmethylated and low methylation loci (beta value < 0.3 on a scale ranging from 0 to 1), with very low methylation dynamics (<10% change, a metric corresponding to 0.1 in beta-value change). Hypermethylation occurred to a greater extent (>50% change, i.e., >0.5 beta change) and mostly impacted medium-/high- methylation sites (beta value > 0.5) (Figure 3). Further analysis of areas spanning multiple CpGs areas consolidated these into 222 DMRs (with methylation differential > 10%), and in high grades, a majority of these were hypermethylated (200 DMRs) with only few hypomethylations (22 DMRs) (Table 3 and Appendix A). Indeed, hypermethylated loci had rather restricted distribution, as 33.8% of these CpGs hit the same gene at least twice. Furthermore, they were distributed preferentially within CpG islands (CGI context) in promoter and intergenic regions (gene context), hinting at enhancer-linked functions. The hypomethylated loci were spread more evenly, with 88.5% hypomethylations located within the gene body of exclusive genes and predominantly in regions outside the CGI (open-sea probes) (Figure 3).

We next conducted unbiased gene ontology analysis on the set of genes associated with the 2099 differentially methylated CpGs. The most enriched biological process ontologies in high-grade versus low-grade meningiomas concerned main functions such as morphogenesis, neurogenesis, and cell differentiation (all FDR < 1 × 10^−13^) (Table 4) and included growth (FDR = 2.80 × 10^−3^), cell leading edge (FDR = 2.89 × 10^−3^), cell cycle (FDR = 2.95 × 10^−3^), and apoptotic process (FDR = 4.97 × 10^−6^). Highly enriched GO terms comprised more specific processes, such as cell morphogenesis involved in neuron differentiation (FDR = 2.81 × 10^−11^, 464/555 genes), positive regulation of transcription by RNA polymerase II (FDR = 9.16 × 10^−10^, 893/1151 genes), regulation of nervous system development (FDR = 3.41 × 10^−9^, 712/894 genes), Rho protein signal transduction (FDR = 3.31 × 10^−5^, 159/191 genes), mitotic cell cycle checkpoint (FDR = 3.31 × 10^−4^, 132/164 genes), regulation of the extrinsic apoptotic pathway (FDR = 4.79 × 10^−4^, 121/149 genes), DNA damage checkpoint (FDR = 9.34 × 10^−3^, 117/142 genes), and cellular response to prostaglandin stimulus (FDR = 0.02, 21/21 genes) (Appendix A).

Among the DMRs and multiple gene hits (Table 3 and Appendix A, Figure 2), notable hypomethylated regions in high-grade meningiomas were linked to cell cycle, cell differentiation, and cell fate genes, such as *SMC4*/*miR16* (four CpGs spanning 1036 bp, 41% average methylation differential (AMD))*, PATJ* (three CpGs spanning 155 bp, 31% AMD), and *TP63* (nine CpGs spanning 650 bp, 23% AMD; coding for the tumor protein p63). DMR hypermethylation in high-grade meningiomas concerned important genes of neural development, such as *CALCB* (CpGs spanning 770 bp, 28% AMD), *PAX6* (DMRs interspersed by 4.6 Kb and associated with alternative transcription start sites: (i) 15 CpGs, 2.76 kb, 27% AMD; and (ii) 9 CpGs, 1.21 kb, 17% AMD)*,* the *PCDH* gene clusters (3 CpGs, 374 bp, 18% AMD), and *WNK2* (8 CpGs, 1169 bp, 25% AMD). There was also a strong enrichment in *SUZ12* (4.8-fold, q-value = 1.33 × 10^−14^) and EZH2 (5.9-fold, q-value = 2.87 × 10^−4^) target regions, both determinants of the polycomb repressor complex 2, which is involved in gene-silencing processes.

### 3.4. Correlation between DNAm and Mitotic Index

CpG methylation levels were significantly correlated with mitotic index (MI) in 891 occurrences for positive correlations (PC) and 2,375 (77%) for negative correlations (NC) (FDR < 0.05, Spearman test; Appendix A, Figure 3). Moreover, with 94.2% very high correlations (|rho| > 2/3, 32/34 CpGs) being negative, DNAm was predominantly negatively correlated with MI. Most NC-linked CpGs were in the medium-methylation category (0.3 > beta value < 0.7) and displayed high methylation dynamics (60–90% variation). Conversely, PC with MI mostly covered methylated CpGs (beta-value > 0.7) with feeble methylation changes (<10%). Overall, these 3,266 correlated loci represented open-sea probes (70% in NC and 65% in PC) and were distributed evenly in proximal promoters, gene bodies, and intergenic contexts (Figure 3).

We obtained further molecular insights after cross-examination of significant loci reported between grade and MI. Intersection of grade-related CpGs with MI-related CpGs resulted in 73 hypermethylations/PC and 263 hypomethylations/NC. No other overlap existed between the four lists, implying the presence of a bidirectional DNAm regulatory mechanism for PC CpGs within CGI and regulatory regions and for NC CpGs outside CGIs in gene bodies (Figure 3). A list of the top correlated hits/genes with MI highlights the biologically relevant methylation candidate markers in meningiomas (Table 5, upper part). These hits were selected based not only on their very high methylation ranges (max–min > 50% and Q3–Q1 > 10%) but also on their gene-wise aggregation in the regulatory elements of the genes (annotations as islands, shores, and shelves within the CGI context, as well as in proximal or distal promoters in the gene context). These included CpGs linked to *SMC4*/*miR16*, p53 effector *CD82*, and probable methyltransferase *METLL24* (seven, two, and two hits, respectively) for NC and CpGs linked to transcription factors *ESRRG*, *PAX9*, and *OTX1* for PC (six, two, and two hits, respectively). Together with other dynamic and island-restricted CpGs, such as *ARHGDIA* (3 hits, NC), *DOK7* (2 hits, NC), *CAPN2* (2 hits, NC), the *PCDH* clusters (10 hits, PC) and *PAX6* (10 hits, PC), these candidates constitute potential proliferative biomarkers associated with disease progression (Appendix A).

### 3.5. DNAm and Ki-67 Labeling Index

CpG methylation levels and Ki-67 labeling index (LI) correlated significantly in 11,532 occurrences, with 4536 PC and 6996 (61%) NC (FDR < 0.05; Appendix A, Figure 3). DNAm was also largely negatively correlated with Ki-67 levels, with 88.4% NC at a very high cutoff (|rho| > 2/3, 38/43 CpGs). Similar to what was observed with MI, the majority of NC-linked CpGs were in the medium (0.3 > beta value < 0.7) methylation category and displayed high (50–90%) methylation dynamics, whereas DNAm PC with Ki-67 expression covered methylated CpGs (beta value > 0.7), mostly displaying moderate methylation variations (<25%). The proportion of CGI and known promoters was also higher in PC than in NC CpGs (Figure 3).

Ki-67 expression was highly correlated with MI (Spearman’s Rho = 0.71, *p*-value = 1.22 × 10^−8^). Surprisingly, less than half of the CpGs correlating with MI also correlated with Ki-67 levels (1185 for NC and 485 for PC). None of the loci in NC with Ki-67 overlapped with those in PC with MI and vice-versa. As with MI, we found no CpG overlap between NC with Ki-67 and hypermethylated sites in high grades, nor for CpGs in PC with Ki-67 and hypomethylated sites in high grades (Figure 3).

A list of the top correlated hits/genes relevant to Ki-67 LI is presented in Table 5 (middle part; same criteria as above with MI), including CpGs linked to *SMC4*/*miR16* and the GTPases *RAB33B* and *VAV2* (9, 2, and 4 hits, respectively) for NC and CpGs linked to long non-coding RNA *CCDC140* and *ESRRG* and neuropeptide *NPY* for PC (10, 7, and 4 hits, respectively). Candidates such as *DOK7* (3 hits, NC), the *PCDH* clusters (32 hits, PC), *PAX6* (15 hits, PC), and *TBR1* (6 hits, PC), among others with dynamic and island-restricted CpGs, constitute potential proliferative biomarkers in meningiomas (Appendix A).

### 3.6. DNAm and MCM6 Labeling Index

CpG methylation levels and MCM6 LI correlated significantly in 4253 occurrences, with 2932 (69%) PC and 1321 NC (FDR < 0.05; Appendix A, Figure 3). In contrast to what was observed with MI and Ki-67 LI, DNAm was generally positively correlated with MCM6 levels, with 70% PC at a very high cutoff (|rho| > 2/3, 7/10 CpGs). NC-linked CpGs were in the low/medium (beta-value < 0.4) methylation category and displayed high (50–90%) dynamics, whereas PC with MCM6 expression covered highly methylated CpGs (beta value > 0.8), most of which displayed moderate methylation variations (<25%). The proportion of island CpGs was also much higher in PC than in NC (Figure 3).

Nevertheless, MCM6 expression correlated positively with MI (Rho = 0.6, *p*-value = 5.31 × 10^−6^) and Ki-67 LI (Rho = 0.69, *p*-value = 7.77 × 10^−8^). Less than one-fifth of the CpGs correlating with MCM6 levels also correlated with MI (321 for NC and 306 for PC), with still no overlap between PC in MI and NC in MCM6 and vice-versa. Conversely, more than a half of the CpGs correlating with MCM6 levels also correlated with Ki-67 levels (856 for NC and 1,337 for PC). Again, no overlap was found between PC in MCM6 and NC in Ki-67 and vice-versa. Finally, as with MI and Ki-67, we found no CpG overlap between those in NC with MCM6 and hypermethylated sites in high grades, nor for CpGs in PC with MCM6 and hypomethylated sites in high grades (Figure 3). Thus, the controlled correlation structure observed between top loci associated with MI and WHO grades held and propagated with Ki-67 and MCM6 markers. The biologically relevant methylation markers based on the top correlated hits with MCM6 LI included CpGs linked with DNA replication inhibitor *GMNN* and ephrin *EFNA1* (a single hit for both) for NC CpGs linked with brain transcription repressor *TBR1* and *PCDH* gene clusters for PC (2 and 27 hits, respectively) (Table 5, lower part). Along with other candidates also associated with WHO grades, such as *SMC4* (three hits, NC) and *PAX6* (four hits, PC), these genes may constitute other interesting progression biomarkers in meningiomas (Appendix A).

### 3.7. DNAm Proliferative Signature in Meningiomas

We did not observe a tight relationship between MI and Ki-67 DNAm markers and even less so with MCM6 levels, as only a small proportion of hits intersected (Figure 3). However, the overlapping structures were conservative between the three correlation experiments. Hence, we next evaluated the dependencies and the complementarities of these measurements (MI, Ki-67, and MCM6 LI) and derived a DNAm proliferation signature in meningiomas. This 310-CpG signature was obtained by cumulating top hits from the three positive (247 intersected CpGs) and three negative (292 intersected CpGs) correlation analyses (Figure 3) and by limiting the results to highly dynamic variables only (standard deviation > 10%; Appendix A). Hierarchical clustering revealed a strongly correlated CpG structure, with an overall progressive sample demethylation according to both proliferation—with a very good adequation with MI—and disease progression, as the signature was also associated with WHO grade (Figure 4). Indeed, benign meningiomas (left side of the heat map) were methylated for a large open-sea CpG set and unmethylated at island CpGs, which constituted one-fifth of the signature. Samples with very low MCM6 LI (<30%) aggregated within this cluster. Grade 3 samples belonged to another cluster (right side of the heat map) of extreme methylation values for these CpGs. This cluster also aggregated samples with very high Ki-67 LI (>70%).

This DNAm signature was validated on an external meningioma dataset (GSE200321) of 60 QC-passed samples of various grades and histological subtypes, which also included invasiveness information (Appendix A). Hierarchical clustering again revealed the same patterns of progressive demethylation at open-sea CpGs across samples with increasing invasive and aggressive properties, accompanied by a strong methylation of island CpGs. Notably, chordoid meningioma is a morphological subtype designated as WHO grade 2 but mostly present with a low proliferation profile comparable to that of grade 1 meningiomas. These results are in line with recent findings suggesting that histology alone may not justify a grade 2 designation for chordoid meningiomas [30].

### 3.8. Associations between DNAm and Survival

We next conducted survival analyses with Cox proportional hazard models to identify specific loci associated with PFS and OS. We first examined the DNAm proliferative signature described above. As expected and independent of WHO grade and age, these 310 CpGs were all associated with survival, albeit better for PFS, with *p*-values in the range of [1.34 × 10^−5^–1.47 × 10^−3^], than for OS, with *p*-values in the range of [1.69 × 10^−3^–2.8 × 10^−2^] (Appendix A; Wald test of multivariate settings).

We then examined genome-wide univariate associations with survival and observed a stronger DNAm association with PFS (69 hits, *p* < 1 × 10^−5^) than with OS (no hits, *p* < 1 × 10^−5^; 54 hits with *p* < 1 × 10^−4^) (Figure 5). Furthermore, PFS was associated massively with chr1q and chr20 loci, with 9.5- and 4.7-fold enrichments against background CpGs, respectively (both *p* < 2.2 × 10^−16^, chi-squared test; hits with *p* < 1 × 10^−4^). No such associations were found with OS. From these top univariate hits, we derived multivariate associations with PFS (Appendix A) and OS (Appendix A). One key conclusion is that for PFS, most loci (48/69) had a positive effect size, meaning that a methylation gain was linked to improved survival. This result echoes our findings based on grade/proliferation: the higher the proliferation (and the higher the grade), the lower the extent of DNAm (Figure 4). This trend was not observed for OS.

## 4. Discussion

DNAm is an innovative tool that is increasingly used for the classification of brain tumors and useful to predict their clinical outcome [29]. In our setup of 48 samples with varied disease progression, all samples were recognized as meningiomas by the main brain tumor classification tool. However, regarding the specific meningioma algorithm, MNGv2.4, and despite our choice of using only freshly frozen tissues, we were still confronted with a high rate of non-classifiable meningiomas (35%). This rate was even higher for meningioma subclassification (46%), which limited its use. Our study unveiled the association between indices of proliferation with disease progression and DNAm and consequently helped to identify a targeted proliferation-relevant DNAm signature comprising only a few hundred CpGs. As this signature could easily be used to score or classify meningioma samples, we propose that such a methylation-based classifier could ameliorate their categorization.

In our attempt to link proliferation/grade/disease outcomes and DNAm, we observed two main dynamics associated with cell proliferation in meningiomas. First, there were few but strong methylations at CGIs located in proximal promoters and enhancers mostly affiliated with neural transcription factors and tumor suppressor genes. One such hit is the developmentally targeted epigenetic silencing by polycomb repressor complex 2; although it is difficult to extrapolate these previous results to meningiomas, this silencing is well-documented in brain cancers [31,32]. Second, there was an extended and progressive demethylation at open sea, mostly within gene bodies. This observation has also been reported in cancers [33,34] although in pediatric brain cancers, the enriched hits were found to be intergenic, with no gene affiliation [6]. This mechanism has been associated with chromosomal instability [35] and malignant progression of lower-grade glioma [36] and was predictive of response to standard chemotherapy in osteosarcoma [37].

Among the epigenetic changes between low- and high-grade meningiomas, we found expected pathways driving unchecked cell division, such as the hypomethylation of genes promoting cell cycle, growth, differentiation, and fate. We also reported dysregulation of regulatory pathways of DNA damage checkpoints and extrinsic apoptosis, along with genes with full driving potential, such as *TP63* [38,39] and *SMC4*. *SMC4* was found to be involved in tumor cell growth, migration, and invasion. It is also correlated with poor prognosis in some cancers. Indeed, its overexpression is suspected to play a role in numerous cancers, such as hepatocellular, colorectal, breast, and endometrial cancer [40,41,42,43,44]. In glioma, *SMC4* overexpression promotes aggressive phenotypes by TGFβ/Smad signaling [45,46,47]. Hence, the hypomethylation of *SMC4* in high-grade meningiomas could likely lead to overexpression, which is consistent with previous data indicating poor prognosis in other tumors. It would therefore be interesting to evaluate its predictive role in meningioma, as it was also highly correlated with all proliferation indices. Additionally, hypermethylated markers in high-grade meningiomas included proximal promoters of *PAX6* and *PCDH* genes, in addition to correlation with proliferation markers. PAX6, a transcription factor playing an important role in the development of the central nervous system, constitutes a good candidate in our study, as in many regulatory loci, it was positively correlated with all proliferative markers. In glioblastoma, a few studies have suggested that a low *PAX6* expression level should be considered prognostically pejorative [48,49,50]. The protocadherin (PCDH) family of proteins plays an important role in neural cell aggregation, cell recognition, and neural development. PCDHs are the “barcodes” of the cell, generating single-cell diversity in the brain [51]. They are encoded by combinations obtained by alternative splicing from the three mapped-in-tandem, multigene *PCDH* α, β, and γ clusters. This splicing is controlled through DNAm states of alternative promoters [52], *PCDH* cluster expression involving CTCF interaction, which can be affected by methylation alterations, leading to long-range epigenetic silencing in cancers [53]. Here, we highlight strong and spread-out PCDH promoter hypermethylations linked to disease progression and cell proliferation markers. We suggest that these aberrant methylations may be of diagnostic value in meningioma. Finally, our findings with respect to the hypermethylation of the tumor suppressor and familial meningioma *WNK2* in grade 2 and 3 meningioma are consistent with previously published data, which suggested epigenetic alterations to be the dominant, grade-specific mechanism of gene inactivation [54,55].

When considering mitotic activity in meningiomas, we found negative correlations with DNAm for *ARHGDIA* and *DOK7* and positive correlations for *ESRRG* and *OTX1* (both correlating with all proliferation indices). *ARHGDIA* codes for a Rho GDP dissociation inhibitor protein with antiapoptotic activity and is involved in cellular processes, such as cell proliferation, cell cycle progression, and cell migration [56]. Its expression is altered in various cancer, such as breast, prostate, and hepatocellular cancer [57,58,59]. In glioma, its level was reported to be correlated with positive prognosis and was considered both an independent predictive marker for OS [56] and an actionable therapeutic target [60]. *DOK7* was also reported to be correlated with the Ki-67 index and may function as a tumor suppressor gene, which is consistent with our findings. However, in breast carcinoma, significant *DOK7* promoter hypomethylation implicated its role in early tumorigenesis [61,62]. In lung cancer, lower *DOK7* expression was associated with lower survival. In in vitro studies, *DOK7* was reported to inhibit proliferation and migration by downregulating the PI3K/AKT/mTOR pathway [63,64]. Similarly, *DOK7* was found to be downregulated in glioma, and its overexpression inhibited both in vitro and in vivo proliferation of glioma cells [65]. ESRRG is a transcriptional activator of *DNMT1* (DNA methyltransferase 1). Its downregulation correlates with poor clinical outcome in gastric cancer, where it acts by suppressing cell growth and tumorigenesis. It also antagonizes Wnt signaling [66]. *ESRRG* promoter hypermethylation was used as a diagnostic and prognostic biomarker in laryngeal squamous cell carcinoma [67]. OTX1 is a transcription factor that is required for proper brain and sensory organ development in mice [68], where it regulates cell cycle progression, with its knockdown leading to diminished neurons but increased astrocytes [69]. Few studies have linked *OTX1* overexpression with tumorigenesis and growth in cancers [70,71], and to the best of our knowledge, it has not been yet reported as an epigenetic marker in brain cancers.

Between DNAm and Ki-67 levels, we report negative correlations for VAV2, a guanine exchange factor playing an important role in angiogenesis. Its recruitment by phosphorylated EPHA2 is critical for EFNA1-induced RAC1 GTPase activation, as well as vascular endothelial cell migration and assembly. Interestingly, we also found EFNA1 DNAm to be correlated with MCM6 expression. VAV2 is overexpressed in numerous cancers [72,73,74], but its dysregulation has not yet been reported in brain cancers. It is implicated in both cutaneous and head and neck squamous cell carcinomas, where it promotes regenerative proliferation [75]. In esophageal squamous cell carcinoma, it is required for DNA repair and radiotherapy resistance [76]. Because it is a Rho guanine exchange factor, VAV2 is an attractive pharmacological target, with druggable catalytic sites and a more restricted tissue distribution pattern than other Rho proteins. *TBR1* DNAm is positively correlated with the Ki-67 and MCM6 indices. TBR1 is a transcriptional repressor involved in multiple aspects of cortical development, including neuronal migration and axonal projection. Recurrent variations of *TRB1* have been described in medulloblastoma, and a high frequency of its copy-number loss has been detected in glioblastoma, suggesting a possible involvement in tumorigenesis or progression [77,78]. To the best of our knowledge, no data are available in the literature with respect to any implication of these genes in meningioma. 

With respect to MCM6 levels, we found negative correlations in *GMNN-* (geminin) and *EFNA1-* linked DNAm. GMNN was shown to inhibit DNA replication by preventing the incorporation of MCM complexes into prereplication complexes (pre-RC) [79,80,81,82,83]. Further evidence implies its participation in the inhibition of the transcriptional activity of a subset of Hox proteins, linking GMNN to proliferative cell cycle control [83]. *EFNA1* codes for the receptor protein-tyrosine kinase Ephrin A1, which is known to mediate developmental events and is involved in migration, repulsion, and adhesion during neuronal, vascular, and epithelial development in the nervous system and in erythropoiesis. In the context of tumor biology, it was shown to be regulated by hypoxia, and its main function involves angiogenesis and tumor neovascularization [84]. The recruitment of VAV2 is critical for EFNA1-induced RAC1 GTPase activation and vascular endothelial cell migration and assembly. 

When correlating proliferative indices (mitotic, Ki-67, and MCM6) with DNAm levels, we found that the positive and negative correlations exclusively occurred in the hyper- and hypomethylated DMRs differentiating WHO grades. This interrelation hints at a tight epigenetic control of cell proliferation. Thus, either on islands or in open-sea regions, non-concerted modification of methylation does not seem to be the key determinant driving cell division in meningioma. Moreover, fewer overlaps than expected were found between hits with our three chosen markers of proliferation, suggesting their complementing diagnostic value. As expected, because these markers are expressed during different stages of the cell cycle with different magnitudes of expression level, Ki-67 was found to perform well with high-grade and highly proliferative meningiomas, whereas MCM6 expression better delineated low-grade, low-proliferation tumors. Overall, the mitotic index was progressively correlated with demethylation in our proliferation signature.

Due to the limited number of samples in our dataset, the survival statistics were capped to a threshold of associations incompatible with genome-wide multitesting corrections. To overcome this, we first focused on the subset of 310 CpGs with proliferation-dependent DNAm and found that all CpGs were associated with PFS or OS. We next looked at genome-wide results and found that PFS associations with DNAm were the strongest, with unexpectedly numerous hits of high significance in broad genomic regions such a chromosomes 1q and 20. Results with PFS also confirmed the directionality of our DNAm proliferation signature, with progressive demethylation associated with worse progression. The lack of such an observation in OS indicates that no high adequation exists between these proliferation markers and occurrence of death; however, this could also be explained by a smaller number of events and the inclusion of deaths due to another pathology (i.e., not specific to OS). OS may be more strongly associated with independent or indirectly linked loci, such as the *MSI1* regulatory region. This gene encodes an RNA binding protein that could be involved in the maintenance of stem cells in the central nervous system and in cell proliferation. In glioblastoma, it was found to promote the expression of stem cell marker CD44 by impairing miRNA function [85]. In these cells, it was shown to be regulated and stabilized by HuR [86], another RNA binding protein and biomarker of interest in meningiomas [87].

In the present study, given the importance of proliferation in the prognosis of meningiomas, we designed an original approach focusing on the correlation between DNAm and cell proliferation. Proliferation is one of the main mechanisms of tumorigenesis and involves a considerable number of pathways shared by multiple types of tumors. However, epigenetic regulation is only a part of the mechanisms by which cells alter their gene expression. Although we do not have evidence of how DNAm changes affect gene expression, DNAm could potentially regulate downstream genes expression, which may contribute to tumorigenesis or progression. To better dissect the underlying molecular pathways of aberrant cellular growth in meningiomas, integrative and multi-omics studies are needed. As a first step, in this pilot mono-omic study integrating molecular and histological indices, we identified a proliferation signature encompassing hundreds of regulated genes, with several candidates serving as potential predictive and prognostic biomarkers or new therapeutic targets. Most notably, *SMC4, DOK7, PAX6, ARHGDIA, ESRRG, VAV2,* and *OTX1*, as well as the three protocadherin gene clusters PCDHα, β, and γ, may be particularly relevant and deserve further preclinical and clinical investigations. 

## 5. Conclusions

In conclusion, this study highlights the crucial role played by epigenetic mechanisms in shaping dysregulated cellular proliferation in meningioma. It provides molecular biomarkers with potential to revamp the current prognostic classification, as well as new druggable targets, adding therapeutic value to clinical management. The reported findings are novel and show the additional value of DNAm evaluation in diagnosis and prognostication for patients with meningioma.

## Figures and Tables

**Figure 1 cancers-14-06227-f001:**
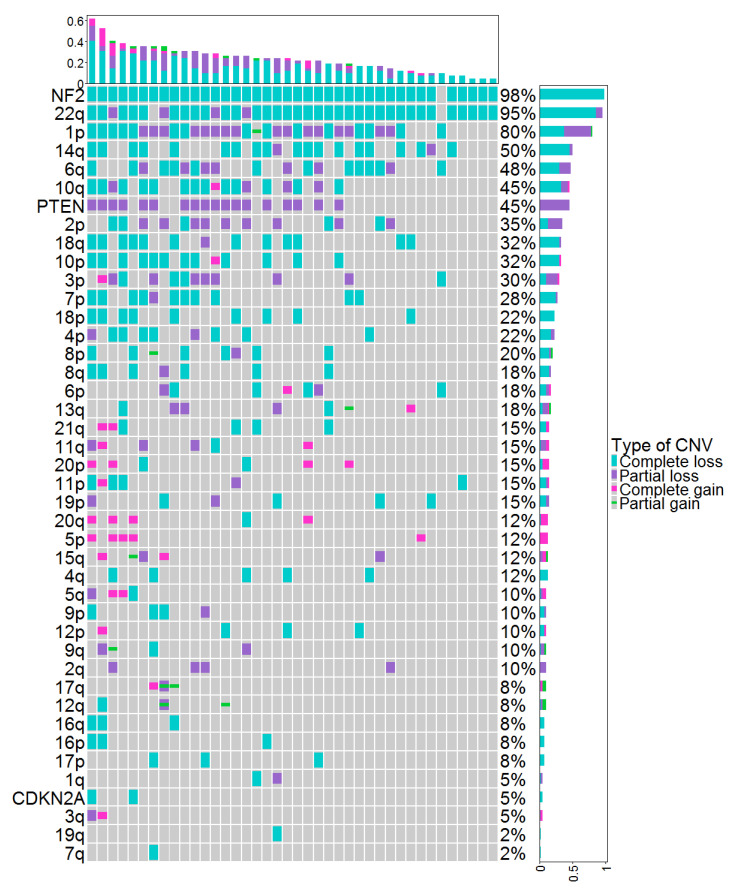
CNVs identified by DNAm in the cohort of 48 meningiomas. Each column represents individual patients ordered from left to right by increasing frequency of genetic alterations. Genetic alterations are ordered on the y axis from top to bottom by decreasing frequency of genetic alterations. CNV: copy-number variation, DNAm: DNA methylation.

**Figure 2 cancers-14-06227-f002:**
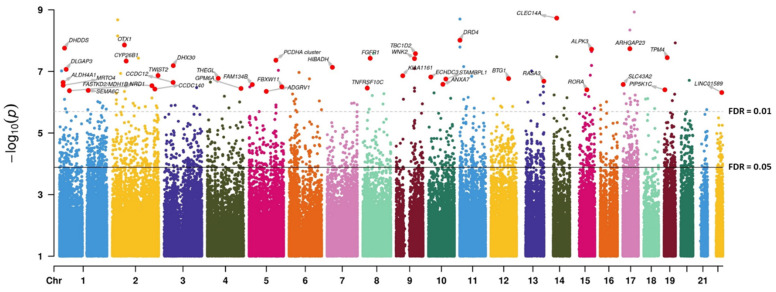
Manhattan plot representing the association between DNAm at the CpG level and high-grade (>1) vs. low-grade meningiomas. The dashed and solid lines indicate FDR thresholds after *p*-value correction for genome-wide multitesting (Benjamini–Hochberg). FDR: false discovery rate.

**Figure 3 cancers-14-06227-f003:**
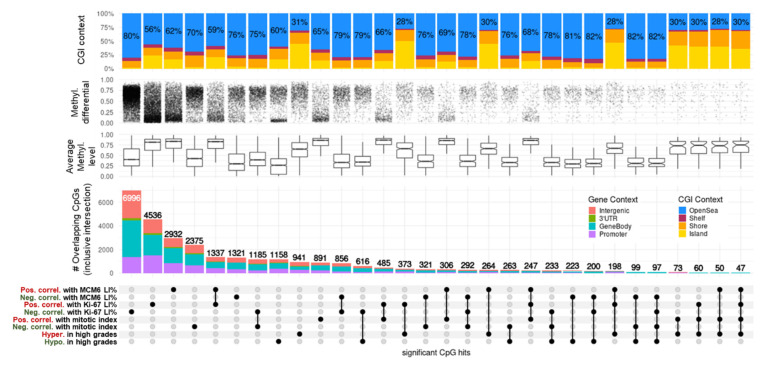
UpSet plot summarizing the overlap between CpGs highly correlated with proliferation indices and grades in meningiomas. Every possible intersection is displayed (30 in total, including the 8 original hit lists shown on the bottom-left side). All intersections are inclusive. CpG regulatory contexts (gene and CGI annotations) are represented as proportions of barplot heights. Proportions of open-sea CpGs are displayed on CGI context bar plots. Average methylation levels are represented as boxplots of means (beta value) for each CpG of the intersected lists. Methylation differentials represent the range of each CpG beta value (max–min) and the densities of changes. DNAm: DNA methylation; methyl: methylation; CGI: CpG island; pos. correl: positive correlation; neg. correl: negative correlation; hypo: hypomethylation; hyper: hypermethylation; LI: labeling index.

**Figure 4 cancers-14-06227-f004:**
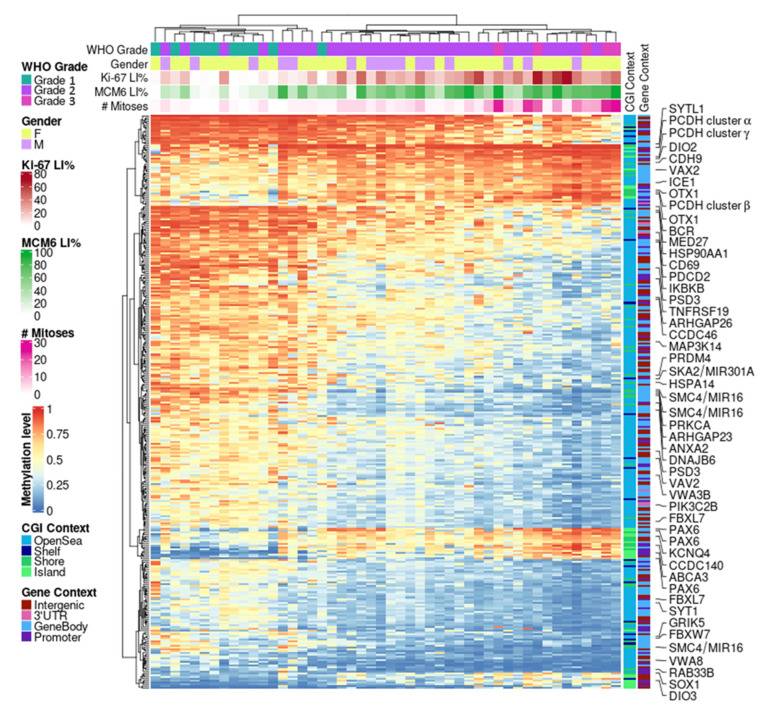
DNAm proliferation signature in meningiomas. Hierarchical clustering heat map depicting 310 CpGs highly correlated between DNAm and three proliferative markers: mitotic index, Ki-67, and MCM6 expressions (metrics: Euclidean distance and complete linkage). CGI: CpG island; DNAm: DNA methylation; LI: labeling index; WHO: World Health Organization.

**Figure 5 cancers-14-06227-f005:**
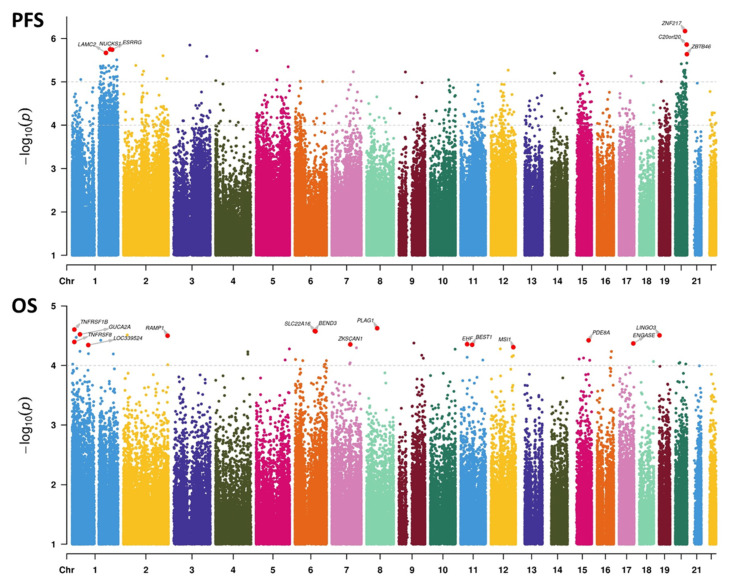
Manhattan plots representing the association between DNAm at the CpG level and survival. Upper panel: associations with progression-free survival (PFS). Lower panel: associations with overall survival (OS). Univariate Cox statistics (Wald test). *p*-value cutoffs running multivariate analyses on selected CpGs are indicated by the topmost dashed lines in each panel. For the PFS panel, the second dashed line represents the threshold for computing hit enrichments over chromosomic regions.

**Table 1 cancers-14-06227-t001:** Clinical and pathological features.

WHO Grade (2021 Classification)	*n* = 48
Grade 1	10 (21%)
Grade 2	33 (69%)
Grade 3	5 (10%)
Age	57 (48; 67)
Sex	
Female	32 (67%)
Male	16 (33%)
Localization	
Skull base	12 (25%)
Convexity	33 (69%)
Ventricular	2 (4%)
Spinal	1 (2%)
Complete resection	
Yes	26 (54%)
No	16 (33%)
Unknown	6 (13%)
Adjuvant chemotherapy	1 (2%)
Adjuvant radiotherapy	22 (46%)
Progression	19 (40%)
Median progression-free survival (months)	39 (16; 55)
Death	9 (19%)
Median overall survival (months)	52 (31; 95)
Ki67 (%)	21 (9; 38)
MCM6 (%)	51 (29; 73)
Mitoses/1.6 mm^2^	2 (1; 6)

Results are presented as absolute values (*n*, %) or median (Q1; Q3). WHO: World Health Organization.

**Table 2 cancers-14-06227-t002:** Molecular features from Sahm et al.’s methylation classifier reports according to the WHO grade.

WHO Grade	Grade 1 (*n* = 10)	Grade 2 (*n* = 33)	Grade 3 (*n* = 5)	Total (*n* = 48)
Methylation class				
Benign	8 (80%)	5 (15%)	0 (0%)	13 (27%)
Intermediate	2 (20%)	12 (36%)	0 (0%)	14 (29%)
Malignant	0 (0%)	2 (6%)	2 (40%)	4 (8%)
No match (calibrated score < 0.9)	0 (0%)	14 (42%)	3 (60%)	17 (35%)
Methylation sub-class				
1	2 (20%)	1 (3%)	0 (0%)	3 (6%)
2	4 (40%)	3 (9%)	0 (0%)	7 (15%)
3	0 (0%)	0 (0%)	0 (0%)	0 (0%)
4	1 (10%)	9 (27%)	0 (0%)	10 (21%)
5	0 (0%)	2 (6%)	0 (0%)	2 (4%)
6	0 (0%)	2 (6%)	2 (40%)	4 (8%)
No match (calibrated score < 0.9)	3 (30%)	16 (48%)	3 (60%)	22 (46%)
*CDKN2A/B* homozygous loss	0 (0%)	0 (0%)	2 (40%)	2 (4%)
*PTEN* loss	0 (0%)	14 (42%)	4 (100%)	18 (38%)
*NF2* loss	5 (50%)	30 (91%)	4 (100%)	39 (81%)

Results are presented as absolute values (*n*, %). WHO: World Health Organization.

**Table 3 cancers-14-06227-t003:** Top differentially methylated regions (DMRs; 15 hypermethylated, 10 hypomethylated) ordered by average methylation differential (Mean Diff.) between high-grade (2, 3) and low-grade (1) meningiomas.

	Linked Gene	Chromosome	# CpGs	FDR	Mean. Diff.	Max. Diff.
Hypermethylated in high grades	CYP26B1	chr2	3	5.21 × 10^−15^	+29%	+42%
	REC8	chr14	13	1.09 × 10^−38^	+29%	+37%
	C2CD4D	chr1	5	6.10 × 10^−13^	+28%	+35%
	KIFC2	chr8	4	2.45 × 10^−14^	+28%	+35%
	CALCB	chr11	5	4.77 × 10^−14^	+28%	+38%
	HEPACAM	chr11	4	1.02 × 10^−16^	+28%	+38%
	DCDC2C	chr2	5	7.55 × 10^−14^	+27%	+41%
	PAX6	chr11	15	7.63 × 10^−23^	+27%	+42%
	SPEG	chr2	6	8.06 × 10^−22^	+27%	+35%
	LTBP4	chr19	5	2.84 × 10^−11^	+26%	+32%
	WNK2	chr9	8	1.11 × 10^−35^	+25%	+43%
	PITX1	chr5	6	5.17 × 10^−14^	+25%	+34%
	KLB	chr4	4	2.85 × 10^−11^	+25%	+33%
	B4GALNT1	chr12	5	1.41 × 10^−14^	+24%	+36%
	IRX1	chr5	8	2.14 × 10^−15^	+24%	+37%
Hypomethylated in high grades	SMC4/miR16	chr3	4	2.30 × 10^−16^	−41%	−46%
	ARHGAP23	chr17	3	6.68 × 10^−24^	−36%	−44%
	PATJ	chr1	3	6.07 × 10^−12^	−31%	−33%
	CACNA1H	chr16	3	3.94 × 10^−14^	−28%	−35%
	THSD4	chr15	3	4.52 × 10^−12^	−25%	−27%
	DNAJB6	chr7	7	2.61 × 10^−13^	−23%	−36%
	TP63	chr3	9	2.38 × 10^−19^	−23%	−42%
	LINC01589	chr22	4	1.30 × 10^−16^	−23%	−32%
	DHX30	chr3	3	1.14 × 10^−13^	−20%	−28%
	RBM47	chr4	11	6.62 × 10^−19^	−19%	−28%

# CpGs: number of CpG probes, FDR: false discovery rate, Mean. Diff. = mean difference (average methylation differential throughout every CpG of the DMR), Max. Diff = maximum difference (single CpG within the DMR showing the maximum methylation differential). Linked genes obtained from overlapping annotations (see Methods). DMRs without linked genes are omitted (full list in Appendix A).

**Table 4 cancers-14-06227-t004:** Top 20 gene ontologies (biological processes) based on the highest significance (FDR) between high-grade (>1) and low-grade meningiomas.

Reference	Ontology Term	N	DE (%)	P.DE	FDR
GO:0009653	anatomical structure morphogenesis	2629	77.7%	1.44 × 10^−27^	3.27 × 10^−23^
GO:0007399	nervous system development	2264	79.1%	6.68 × 10^−27^	7.60 × 10^−23^
GO:0048856	anatomical structure development	5697	73.0%	3.36 × 10^−26^	2.55 × 10^−22^
GO:0032502	developmental process	6073	72.6%	1.65 × 10^−25^	9.36 × 10^−22^
GO:0016043	cellular component organization	6081	73.2%	5.20 × 10^−25^	2.37 × 10^−21^
GO:0007275	multicellular organism development	5227	73.1%	2.03 × 10^−24^	7.71 × 10^−21^
GO:0071840	cellular component organization or biogenesis	6261	72.9%	3.91 × 10^−24^	1.11 × 10^−20^
GO:0048518	positive regulation of biological process	5830	72.6%	2.35 × 10^−23^	5.94 × 10^−20^
GO:0048522	positive regulation of cellular process	5143	73.1%	5.74 × 10^−23^	1.31 × 10^−19^
GO:0048731	system development	4689	73.2%	3.42 × 10^−22^	7.08 × 10^−19^
GO:0048468	cell development	2096	77.8%	2.10 × 10^−20^	3.97 × 10^−17^
GO:0048869	cellular developmental process	4222	73.0%	1.74 × 10^−19^	3.05 × 10^−16^
GO:0022008	neurogenesis	1563	79.5%	3.76 × 10^−19^	6.11 × 10^−16^
GO:0009893	positive regulation of metabolic process	3450	73.9%	4.98 × 10^−19^	7.56 × 10^−16^
GO:0048699	generation of neurons	1465	79.7%	1.95 × 10^−18^	2.77 × 10^−15^
GO:0010604	positive regulation of macromolecular metabolic process	3193	74.1%	3.98 × 10^−18^	5.33 × 10^−15^
GO:0030182	neuron differentiation	1310	80.4%	5.16 × 10^−18^	6.52 × 10^−15^
GO:0030154	cell differentiation	4040	72.7%	1.28 × 10^−17^	1.53 × 10^−14^
GO:0031325	positive regulation of cellular metabolic process	3166	74.0%	1.79 × 10^−17^	2.04 × 10^−14^
GO:0010646	regulation of cell communication	3381	73.9%	2.44 × 10^−17^	2.64 × 10^−14^

GO: gene ontology; N: number of genes in the GO category; DE: percentage of differentially methylated genes (corrected for representation bias and normalized against background) among the GO category; P.DE: over-representation *p*-value; FDR: false discovery rate.

**Table 5 cancers-14-06227-t005:** Top correlated CpGs between DNAm levels (beta values) and mitotic indices, Ki-67 labeling indices, and MCM6 labeling indices in regulatory region contexts.

		CpG	Gene (# Hits)	Chr.	Rho	*p*-Value	DNAm Median Level (Beta-Value)	DNAm Diff. Range (%)
Mitotic index	negative	cg21942721	ASB4 (1)	chr7	−0.71	1.27 × 10^−8^	High (0.84)	60 (26)
		cg18568061	PTRF (1)	chr17	−0.69	4.94 × 10^−8^	Medium (0.37)	63 (21)
		cg01764105	SMC4/miR16 (7)	chr3	−0.69	7.46 × 10^−8^	Low (0.08)	76 (21)
		cg17605814	CD82 (2)	chr11	−0.68	1.22 × 10^−7^	Medium (0.57)	75 (25)
		cg22624818	SDPR (2)	chr2	−0.68	1.34 × 10^−7^	High (0.80)	66 (17)
		cg16166651	DEPDC1 (3)	chr1	−0.67	1.53 × 10^−7^	Low (0.16)	62 (12)
		cg06003566	METTL24 (2)	chr6	−0.67	2.19 × 10^−7^	High (0.86)	79 (14)
	positive	cg25588576	MIR7641-2 (1)	chr14	0.66	3.52 × 10^−7^	High (0.79)	69 (31)
		cg21784383	ESRRG (6)	chr1	0.65	5.83 × 10^−7^	Low (0.18)	71 (27)
		cg18361098	PAX9 (2)	chr14	0.62	2.80 × 10^−6^	Low (0.10)	80 (43)
		cg10640333	OTX1 (2)	chr2	0.61	4.33 × 10^−6^	High (0.76)	90 (20)
		cg13244312	TTC9 (1)	chr14	0.61	4.81 × 10^−6^	Medium (0.58)	55 (19)
Ki-67 LI%	negative	cg01464849	SMC4/miR16 (9)	chr3	−0.73	3.33 × 10^−9^	Low (0.26)	84 (58)
		cg18943088	IQCJ-SCHIP1 (16)	chr3	−0.71	1.78 × 10^−8^	Low (0.30)	83 (42)
		cg22800629	RAB33B (2)	chr4	−0.69	7.13 × 10^−8^	Low (0.09)	53 (15)
		cg17253087	HIPK3 (1)	chr11	−0.68	7.97 × 10^−8^	Medium (0.55)	78 (45)
		cg11629830	IQGAP2 (3)	chr5	−0.68	1.04 × 10^−7^	Medium (0.69)	68 (36)
		cg13944632	VAV2 (4)	chr9	−0.66	2.97 × 10^−7^	Medium (0.37)	75 (15)
	positive	cg03126579	ZFR2 (1)	chr19	0.71	1.27 × 10^−8^	Medium (0.65)	93 (37)
		cg10269365	CCDC140 (10)	chr2	0.69	6.92 × 10^−8^	Medium (0.55)	84 (37)
		cg08139247	CLEC14A (5)	chr14	0.68	9.42 × 10^−8^	Medium (0.53)	83 (48)
		cg21784383	ESRRG (7)	chr1	0.66	3.34 × 10^−7^	Low (0.18)	71 (27)
		cg26418900	NPY (4)	chr7	0.66	3.34 × 10^−7^	Medium (0.40)	87 (32)
		cg10640333	OTX1 (4)	chr2	0.65	5.83 × 10^−7^	High (0.76)	90 (20)
MCM6 LI%	negative	cg04570316	GMNN (1)	chr6	−0.62	2.45 × 10^−6^	High (0.89)	71 (11)
		cg09130952	CCDC108 (1)	chr2	−0.62	2.52 × 10^−6^	High (0.71)	68 (25)
		cg16959792	SLC50A1/EFNA1 (1)	chr1	−0.62	2.57 × 10^−6^	Low (0.23)	39 (18)
		cg24310126	FLJ46361 (1)	chr10	−0.62	2.99 × 10^−6^	High (0.85)	78 (16)
		cg03805253	CACNA1G (1)	chr17	−0.62	3.05 × 10^−6^	Medium (0.68)	61 (19)
		cg10422455	MRGPRX2 (1)	chr11	−0.61	3.69 × 10^−6^	Medium (0.66)	82 (33)
	positive	cg03126579	ZFR2 (1)	chr19	0.68	8.32 × 10^−8^	Medium (0.65)	93 (37)
		cg03552103	SEPT10/ANKRD57 (1)	chr2	0.67	1.70 × 10^−7^	Medium (0.69)	50 (13)
		cg15415136	ZNF540 (2)	chr19	0.66	3.60 × 10^−7^	Low (0.15)	56 (10)
		cg06488443	TBR1 (2)	chr2	0.66	3.93 × 10^−7^	Low (0.29)	61 (22)
		cg05008496	SSPN (1)	chr12	0.64	1.20 × 10^−6^	Medium (0.56)	52 (22)
		cg22151446	PCDHabg clusters (27)	chr5	0.63	1.61 × 10^−6^	Medium (0.31)	66 (38)
		cg12052661	CACNA1B (2)	chr9	0.63	1.72 × 10^−6^	Medium (0.39)	52 (23)

DNAm: DNA methylation; LI: labeling index; # hits: total number of significant CpGs in the same regulatory region; Chr: chromosome; diff: differential. Low (methylation beta value ≤ 0.3), medium (beta value > 0.3 and <0.7), and high (beta value ≥ 0.7) median levels. Differential ranges in max–min % (Q3–Q1%). Spearman’s Rho correlations. Full lists and metrics in Appendix A.

## Data Availability

Publicly available datasets were analyzed in this study. This data can be found here: https://www.ncbi.nlm.nih.gov/geo/query/acc.cgi?acc=GSE200321. The data generated and presented in this study are available upon request from the corresponding author. The data are not publicly available due to privacy restrictions.

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
