# Peer review of "Correlation between DNA Methylation and Cell Proliferation Identifies New Candidate Predictive Markers in Meningioma"

_cancers, 2022, doi:10.3390/cancers14246227_

Round 1

Reviewer 1 Report

In the manuscript “Correlation between DNA methylation and cell proliferation identifies new candidate predictive markers in meningioma”, HERGALANT et al. generated global DNA methylation profiling of 48 meningioma tumor tissues using Infinium MethylationEPIC microarray. By comparing the high-grade group with low-grade group, they identified differentially methylated CpGs and regions, which have potential clinical value. They also demonstrated the correlation between DNA methylation and cell proliferation. The finding is interesting, however there are some questions need to be addressed.

Major comments:

1. To identify the differentially methylated CpGs between WHO grades, the authors only filtered the CpGs by FDR < 0.05, but the absolute methylation change (beta value change) should also be considered. Technically, the beta value change should be significantly greater than error range of EPIC array assay. For example, lines 234-235, “hypomethylation occurred at already unmethylated to lowly methylated loci (beta-value < 0.3) with very low methylation dynamics (< 10% change).”, the beta value change (methylation dynamics) is less than 0.03 (0.3 * 10%). This small change is in the technical error range. As predictive markers, the beta value change should be big enough to have great sensitivity. The authors could run unsupervised clustering heatmap and principal component analysis to check if their differentially methylated CpGs or DMRs could separate high-grade and low-grade groups well, which is very important for predictive markers.

2. Sample size is relatively small. The authors should consider validating their results in public datasets if there is any. In addition, the authors should consider other methods (e.g., bisulfite target sequencing) to validate the EPIC array data.

Specific comments:

1. What is the purposes of calculate the ‘Methylation differentials’ in figure 3, ‘Mean diff’ and ‘Max diff’ in table 3, and ‘DNAm diff. range (%)’ in table 5? Do they have specific meaning?

2. How many genes are associated with the 2099 differentially methylated CpGs? For table 4, please provide gene numbers in this list for each GO term.

3. Table 5, the values of ‘DNAm median level’ should be shown, instead of only the ranks.

4. Again, lines 319-320, what was the reason that the authors chose the ‘high methylation range’ hits? The ‘high methylation range’ means the methylation level is highly variable between samples, which could lead to reduced predictive sensitivity and decrease the value of these hits as predictive markers. Same question to line 395.

5. Figure 4 indicates the methylation profiling of those 310 CpGs could distinguish low grade (grade 1) and high grade (grade 2 and 3) groups, which is very promising, although with several exception. I wonder if the 310 CpGs overlapped with the 2099 CpGs (line 231). The author could further narrow down the list to find the best CpGs signature to predict tumor grade, for example, the CpGs with great methylation difference (like |beta value change| > 0.2 or other cutoff) between low-grade and high-grade groups.

6. In the discussion, the authors talked a lot about the relationship between the DNA methylation associated genes and tumorigenesis. However, they don’t have evidence supporting the DNA methylation change could really affect gene expression in this study. So, a statement, like “Although we don’t have evidence of how the DNA methylation changes affect gene expression in this study, the DNA methylation could potentially regulate the genes expression, which may contribute to tumorigenesis or progression”, should be added.

Author Response

Reviewer1

In the manuscript “Correlation between DNA methylation and cell proliferation identifies new candidate predictive markers in meningioma”, HERGALANT et al. generated global DNA methylation profiling of 48 meningioma tumor tissues using Infinium MethylationEPIC microarray. By comparing the high-grade group with low-grade group, they identified differentially methylated CpGs and regions, which have potential clinical value. They also demonstrated the correlation between DNA methylation and cell proliferation. The finding is interesting, however there are some questions need to be addressed.

We thank the reviewer for his interest, his time, and the accurate description of our work.

Major comments:

  1. To identify the differentially methylated CpGs between WHO grades, the authors only filtered the CpGs by FDR < 0.05, but the absolute methylation change (beta value change) should also be considered. Technically, the beta value change should be significantly greater than error range of EPIC array assay. For example, lines 234-235, “hypomethylation occurred at already unmethylated to lowly methylated loci (beta-value < 0.3) with very low methylation dynamics (< 10% change).”, the beta value change (methylation dynamics) is less than 0.03 (0.3 * 10%). This small change is in the technical error range. As predictive markers, the beta value change should be big enough to have great sensitivity. The authors could run unsupervised clustering heatmap and principal component analysis to check if their differentially methylated CpGs or DMRs could separate high-grade and low-grade groups well, which is very important for predictive markers.

Yes, technically the beta-value change should also be considered for filtering top CpGs between low and high grade meningiomas. We initially did it but finally decided to simplify the report as nearly every CpG passed the statistical power threshold of 75%, which corresponds to about 5% methylation change for 48 samples (Mansell et al., BMC Genomics, 2019, PMID: 31088362). Plus, Illumina methylation microarray have been proven to deliver precise, robust, and highly reproducible results (Pidsley et al., Genome Biology, 2016, PMID: 27717381). To give the reviewer an idea, of the total 2099  CpGs with FDR < 0.05, 2064 (98.3%) presented with beta-value change > 1%, 1752 (83.5%) had a methylation differential > 5%, and 1641 (78.2%) showed > 10% change (added page 7, lines 242-244). So the vast majority of CpGs had a methylation change above the technical error limit. These CpGs are clearly driving the separation we observe between low (1) and high (2+3) grade meningioma, as demonstrated by the PCA below (Figure A, 45.9% variance observed on the first component).

We fear that the misunderstanding of our presentation of methylation dynamics comes from a lack of clarity of how beta-values (ranging from 0 to 1, with low methylation at 0-0.3, medium methylation at 0.3-0.7, and high methylation at 0.7-1) and methylation differences (expressed in %change, so for example 1% to 10% change really mean 0.01 to 0.1 in terms of beta-value, and not 10% of 0.3 = 0.03 as the reviewer understood) are introduced in the Results section. Consequently, we added an explanation in the Methods section (page 4, lines 176-179 and 185-190), and highlights in Results  (page 8, lines 246-248) to improve clarity in the way beta-values and methylation differentials/changes/ranges are presented in the article.

Figure A. PCA of the top 2099 CpGs from the differential analysis between grade 1 and grade 2+3 meningiomas.

  1. Sample size is relatively small. The authors should consider validating their results in public datasets if there is any. In addition, the authors should consider other methods (e.g., bisulfite target sequencing) to validate the EPIC array data.

To our knowledge, the only methylation-wide dataset available/usable and with useful information is relatively new (10.1186/s40478-022-01362-3). The dataset was downloaded from the Gene Expression Omnibus database (GSE200321; https://www.ncbi.nlm.nih.gov/geo/query/acc.cgi?acc=GSE200321) along with all phenotypic variables that can be used within our context of proliferative behaviour (grade and histology). Samples underwent the same quality control (QC), normalization and imputation steps as described in Methods for our dataset. From the original 64 samples available, 60 (6 grade 1 (5 meningothelial and 1 transitional), 50 grade 2 (10 chordoid and 27 atypical, and 13 atypical and invasive), and 4 anaplastic grade 3) passed the QC step and were retained in the final dataset. No data related to mitotic or Ki-67 labelling indices was available.

In our opinion, the best way to validate the EPIC array data would be to test the final 310-CpG proliferative signature in this external dataset (Figure B below). Indeed, for this specific signature we again observe a general demethylation of the same CpGs across samples of increasing invasive properties, accompanied by a strong methylation of a few island-related CpGs. Of note, chordoid meningioma is a histological/morphological subtype designated as WHO grade 2. However, one should bear in mind that histology alone may not allow to accurately predict the prognosis, and from a molecular point of view, most chordoid meningiomas are often reconsidered as grade 1. Conversely, the only one transitional (WHO grade 1) meningioma at our disposal features a more proliferative profile than the other grade 1 tumors. Unsurprisingly, grade 3 (anaplastic) tumors again demonstrate the most extreme methylation profiles.

Given these new findings, we would like to thank the reviewer for this suggestion as it improves the value of our findings. This new figure B was integrated in the manuscript as supplemental Figure S1 (Appendix A) and we now reported the main results at the end of the ‘DNAm proliferative signature in meningiomas’ section (page 4, lines 418-426).

Figure B. Validation of the proliferative signature on an external methylation dataset (https://www.ncbi.nlm.nih.gov/geo/query/acc.cgi?acc=GSE200321; from Daoud et al., Acta Neuropathol. Commun., 2022, PMID: 35440040). Hierachical clustering against the 310 CpGs and 60 samples of various meningioma grades and histological types.

Specific comments:

  1. What is the purposes of calculate the ‘Methylation differentials’ in figure 3, ‘Mean diff’ and ‘Max diff’ in table 3, and ‘DNAm diff. range (%)’ in table 5? Do they have specific meaning?

Yes, as stated by the reviewer and by us in the first major comment, methylation changes are very important and must pass a certain threshold to be considered as valid differences. Step by step, in our analysis of DNAm and proliferation indices, we endeavoured to identify the most authentic differential and the most correlated CpGs presenting high methylation dynamics. Thus, we propose here that to be considered as valid therapeutic targets and/or prognosis, DNA methylation markers of requires to be 1) differential, 2) correlated, and 3) with a high enough range of methylation changes.

In Figure 3, the ‘methylation differential’ (ranging from 0% to 100% here) panel displays the density of CpGs (from each intersection list) present at this level of methylation change. For example, the 4th list (neg correl with mitotic index) is composed of 2375 CpGs that nearly all have about 70% or more methylation dynamic - this information indicates that there is a large spread (great difference) between the minimum and the maximum observed methylation values across the 48 samples. But these are anyway negatively correlated with mitotic index, so when a sample have low methylation, the mitotic index is high and vice-versa. These CpGs are consequently very informative and important. In Layman’s terms, the beta-value is really changing with mitotic index, not stagnant. Correlation alone is not enough to define this “range of change”. We believe this variable is an important one to be displayed, we thus better explain its significance in the Figure 3 legend (added page 9).

In Table 3 we are presenting DMRs, which are composed of at least 3 CpGs each. ‘Mean diff’ represents the averaged difference between all CpGs of each group in the region. ‘Max diff’ represents the maximum observed difference for a given CpG within the region, i.e. the difference between the lowest sample in one group and the highest in the other is calculated for every CpG of the DMR, then the max one is reported. We amended Table 3 legend to clarify these points (page 9).

In Table 5 (page 12), ‘DNAm diff range%’ have the same meaning as ‘Methylation differentials’ in Figure 3. In the legend, we explain: “Differential ranges in Max-Min % (Q3-Q1 %)”. Here we give the extent of methylation change across the 48 samples, reporting the maximum range and the inter-quartile range. These are informative variables to estimate the value of the CpG as a good marker, the correlative statistics alone being too evasive for this purpose.

  1. How many genes are associated with the 2099 differentially methylated CpGs? For table 4, please provide gene numbers in this list for each GO term.

From the annotation with the hg38 reference genome, we can link 1708 genes with hypomethylated CpGs and 996 genes with hypermethylated CpGs, for a total of 2704 unique genes. But many ncRNA or unknown/unprocessed are not included in these annotations, many hits can be linked to the same gene(s) or have many possible genes to link with. As such, functionally annotating raw gene lists coming from methylation studies gives biased results. Unbiased methods such the ones proposed in the R missMethyl package (Phipson et al., Bioinformatics, 2015, PMID: 26424855) have been developed for this very purpose. They correct for over/under-representation of genes in each statistically identified region and normalize against background internally and report a corrected number of differential genes for each functional term (DE). Here we chose to report this number as percentage (DE%) of genes in the GO category (N), which we believe is easier to evaluate than normalized numbers. For example, in the 1st line of Table 4, 77,7% of the 2629 genes have been linked from our top sites by missMethyl. This corresponds to 2041.6 genes but this is a normalized number that should not be interpreted as raw. The complete Table S2 which accompanies Table 3 is already providing this information. We have amended Table 3 legend to reflect this (page 9).

  1. Table 5, the values of ‘DNAm median level’ should be shown, instead of only the ranks.

Changes have been made accordingly in Table 5 (page 12). They can also be found in Tables S3, S4 and S5 along with other precise metrics.

  1. Again, lines 319-320, what was the reason that the authors chose the ‘high methylation range’ hits? The ‘high methylation range’ means the methylation level is highly variable between samples, which could lead to reduced predictive sensitivity and decrease the value of these hits as predictive markers. Same question to line 395.

As explained in answers to major comment 1 and specific comment 1, the CpGs are already highly differential and/or highly correlated, in the statistical sense of the term. But this does not mean that the methylation range, or change, is very high. A good correlation could occur with very low methylation change between the min and the max beta-value. In cancer genomics, there is a need to pinpoint the real, informative, biologically relevant variables among an important number of changes and a large amount of irrelevant noise. We select/highlight these CpGs that have high methylation range or dynamics for this purpose. As these CpGs were already selected statistically for their sensitivity, we add to their value as predictive markers by focusing on those with medium to high dynamics. This is further demonstrated in Figure 4, where we can clearly see the that highly correlated CpGs (with proliferation indices) that also have high methylation ranges increment or decrement gradually along samples. Additionally, this “cutoff“ helps us deal with the lack of power to detect very low methylation changes as we would require a design with hundreds of samples to do so.

  1. Figure 4 indicates the methylation profiling of those 310 CpGs could distinguish low grade (grade 1) and high grade (grade 2 and 3) groups, which is very promising, although with several exception. I wonder if the 310 CpGs overlapped with the 2099 CpGs (line 231). The author could further narrow down the list to find the best CpGs signature to predict tumor grade, for example, the CpGs with great methylation difference (like |beta value change| > 0.2 or other cutoff) between low-grade and high-grade groups.

The primary focus of our study was to find and describe markers correlated with proliferation in meningioma. Associations with WHO grade (which have already been reported by other groups), although important, represent a useful comparative tool and a secondary aim in this context. WHO stratification of meningiomas is far from perfect because based only on histological features and the relatively subjective observations of pathologists. It is currently being challenged by many scientists and clinicians who increasingly report new molecular markers and classifiers that greatly improve our understanding of the histological subtypes, but also help explain the progression, aggressiveness, and recurrence of the disease. It is expected that a proliferative signature such as the one presented here will contain some grade discrepancies and imperfections. As stated above, histology alone is insufficient to characterize the grade of meningioma tumors, so one of our aim was to help improve this. This is the reason why we did not choose to further overlap our multi-overlapping list with WHO grade CpGs.

Anyway, to give the reviewer an idea of the extent of such an overlap, Figure 3 displays 97 hypomethylated + 47 hypermethylated CpGs in high grades, respectively. By further reducing these 144 CpGs to the ones that show high variations, as proposed, we obtain 123 CpGs. This is less than half of our 310-CpG signature and illustrates well the fact that there is a clear link between WHO grade and proliferation.

  1. In the discussion, the authors talked a lot about the relationship between the DNA methylation associated genes and tumorigenesis. However, they don’t have evidence supporting the DNA methylation change could really affect gene expression in this study. So, a statement, like “Although we don’t have evidence of how the DNA methylation changes affect gene expression in this study, the DNA methylation could potentially regulate the genes expression, which may contribute to tumorigenesis or progression”, should be added.

Thank you for this comment. We have made the change accordingly in the end of the discussion (page 19, lines 602-603).

We hope we have clarified things for the reviewer, and we thank him again for helping us improve the overall understanding and quality of the manuscript for the readership.

Reviewer 2 Report

General:

This study further evaluates the value of methylation analysis in patients with meningioma. They propose several new targets for further evaluation considering the diagnosis and prognosis of meningioma patients. In addition, they place these markers in the context of targets for treatment. The paper is original and clinically relevant. Writing is clearly organized and compact.

Title & Abstract:
Title: Adequate

Simple summary: Adequately and clearly describes the research performed.

Abstract: Adequate, describes the findings well. 
Keywords: Adequate.

Introduction:

There is a clear reasoning within the introduction with a concise overview of presently available knowledge.

Materials and Methods:
The timeframes, population, and sample size, as well as outcome measures, are clearly described. Higher grade meningioma was the focus of the study, with grade 1 meningiomas as the control group.

Techniques are sufficiently described and allow the reproduction of the study. Statistical analysis is generally described appropriately. Described, although some elements are missing. Ethics approval is described appropriately.

1.     Overall and progression-free survival is calculated from the date of surgery. To my knowledge, grade 1 meningiomas are generally not operated on at first diagnosis with imaging. Could the survival time be affected by the authors’ choice to calculate survival from the time of operation instead of the time of radiological diagnosis?

2.     The mitotic count was blindly evaluated by two pathologists and the average count was calculated and displayed. How well were the individual counts of a patient’s mitotic count correlated between the two pathologists? Could authors comment on this, please? Were there any definitions of outliers, and if these occurred, how were these handled? 

3.     Authors used the original pathology report as the basis for diagnosis, as treatment was based on that. This is a valid choice, although it would be of interest to have the two pathologists had reviewed the samples to evaluate if there would have been a consensus diagnosis with the original pathologist. Have authors done this, and would they consider this a valuable addition to their manuscript?

4.     To me, it is not 100% clear if the authors checked if the data fulfilled the proportional hazards assumptions. Could the authors state this in the paper?

5.     Authors state that in their multivariate model, they included age and grade as covariates. What did the authors do with treatments such as radiotherapy and/or chemo? Might these modalities have affected (progression-free) survival? 

Results:
In general, the results are clearly described in detail. The figures and tables are clear. 

1.     The manuscript could benefit from a supplementary table describing the clinical treatments of each patient. Were patients treated with other modalities prior to surgery? Was the resection complete? What were the treatments after surgery?

2.     In table 1: the authors included 1 patient with a spinal cord meningioma. I wonder whether such is appropriate in this study as these meningiomas might have a very different clinical course than those within the brain. Could authors comment on this?

3.     If treatments were given prior to surgery (eg the source of tumor material for methylation analysis), could the methylation profiles be affected by the prior treatments?

Discussion and conclusions

The discussion addresses the main question of the manuscript adequately and is in line with the obtained results. Authors attempt to explain their results in relation to other cancer types. New findings and limitations of the study are described in detail. Many comparisons are made with data from primary brain tumors, e.g. glioma/glioblastoma. I feel that this comparison is not totally straightforward. Some caution in this comparison would, in my opinion, be unjustified as they have a very different origin.

The authors are modest in their conclusions; their findings are novel and again show the additional value of methylation evaluation in the diagnosis and prognostication for patients with meningioma.

Author Response

Reviewer2

This study further evaluates the value of methylation analysis in patients with meningioma. They propose several new targets for further evaluation considering the diagnosis and prognosis of meningioma patients. In addition, they place these markers in the context of targets for treatment. The paper is original and clinically relevant. Writing is clearly organized and compact.

We thank the reviewer for the interest and the time spent for the valuable evaluation of our work throughout his comments.

Materials and Methods:

  1. Overall and progression-free survival is calculated from the date of surgery. To my knowledge, grade 1 meningiomas are generally not operated on at first diagnosis with imaging. Could the survival time be affected by the authors’ choice to calculate survival from the time of operation instead of the time of radiological diagnosis?

Indeed, survival was calculated from the date of surgery, similarly to various previous study, and reflect the tumor aggressivity, i.e. the propension to recur / progress after the first surgical treatment ; it could potentially have an impact on the time-to-progression, but not on the status (event vs. censure). Moreover, as the date of first symptoms and/or imaging was not available for all the patients (added in methods, page 3, lines 131-132), it was not possible to use it for our study.

  1. The mitotic count was blindly evaluated by two pathologists and the average count was calculated and displayed. How well were the individual counts of a patient’s mitotic count correlated between the two pathologists? Could authors comment on this, please? Were there any definitions of outliers, and if these occurred, how were these handled?

The agreement was evaluated with the intraclass correlation coefficient (added in methods, page 3, lines 138-139), showing a good inter-observer agreement (0.75) (added in results, page 5, lines 211-213). There was no definition of outliers: a mean value of the two counts was calculated.

  1. Authors used the original pathology report as the basis for diagnosis, as treatment was based on that. This is a valid choice, although it would be of interest to have the two pathologists had reviewed the samples to evaluate if there would have been a consensus diagnosis with the original pathologist. Have authors done this, and would they consider this a valuable addition to their manuscript?

In comparison with the original pathology report, the grade has changed in only one case: this is explained by the detection of a homozygous deletion of CDKN2A/B, leading to a reassessment from grade 2 to 3 (WHO 2021 criteria) (added in results, page 5, lines 208-210).

  1. To me, it is not 100% clear if the authors checked if the data fulfilled the proportional hazards assumptions. Could the authors state this in the paper?

We thank the reviewer for this remark. We have to emphasize that this check must be made for every single test. As we do these analyses methylome-wide, it is impractical to report the statistics for hundreds of thousands of CpG.

So, in univariate analyses, for each CpG, proportional hazard assumption was tested on the fitted model with the cox.zph function included in the R survival package. Two statistics were considered, the p-value of non-random distribution for the methylation variable (CpG), and the global p-value for the model. If both p were found > 0.05, the CpG was retained for further multivariate evaluation. As only and handful of significant hits was conserved for multivariate analyses, we did not recheck proportional hazard assumptions for both age and sex as we did not consider the statistical significance of these two covariates but only that of the methylation covariate after correcting for age and sex.

Accordingly, we have included the required changes in the survival analyses paragraph in Methods (page 5, lines 188-193).

  1. Authors state that in their multivariate model, they included age and grade as covariates. What did the authors do with treatments such as radiotherapy and/or chemo? Might these modalities have affected (progression-free) survival?

In fact, these adjuvant treatments are closely related to other parameters, such as WHO grade (i.e., grade 2 or 3), the extent of surgery (incomplete resection) or the occurrence of a progression. Thus, in this retrospective unrandomized study, the inclusion of these parameters in survival analyses would probably induce statistical biases.

Results:

  1. The manuscript could benefit from a supplementary table describing the clinical treatments of each patient. Were patients treated with other modalities prior to surgery? Was the resection complete? What were the treatments after surgery ?

No neoadjuvant treatment was delivered to the patients included in this cohort.

The quality of resection and adjuvant treatment are detailed in a new supplementary table S1. After verification of the adjuvant treatment, only one patient had chemotherapy (instead of two, confusion with bevacizumab in one case which not strictly chemotherapy). Additionally, few other patients had also treatment with bevacizumab after progression (added in results page 5, lines 212-215).

  1. In table 1: the authors included 1 patient with a spinal cord meningioma. I wonder whether such is appropriate in this study as these meningiomas might have a very different clinical course than those within the brain. Could authors comment on this?

This case was an atypical meningioma, without histological specificity (and notably no clear cell features), with a classical subdural extramedullary localization, and without recurrence. Hence this case seems not to have a different clinical course.   

  1. If treatments were given prior to surgery (eg the source of tumor material for methylation analysis), could the methylation profiles be affected by the prior treatments?

No neoadjuvant treatment was delivered to the patients included in this cohort (added page 5, line 212).

Discussion and conclusions

Many comparisons are made with data from primary brain tumors, e.g. glioma/glioblastoma. I feel that this comparison is not totally straightforward. Some caution in this comparison would, in my opinion, be unjustified as they have a very different origin.

Indeed, the comparison with glioma, as well as other extra-CNS tumors, is debatable, and the results obtained in these tumors may not be extrapolated to meningioma. We added some cautions in the discussion section (page 17, lines 485-486).

The authors are modest in their conclusions; their findings are novel and again show the additional value of methylation evaluation in the diagnosis and prognostication for patients with meningioma.

Thank you for your comments. Following this suggestion, we have made this addition in the conclusion section (page 19, lines 626-628).

Round 2

Reviewer 1 Report

The authors have done extra analysis and detailed clarification to answer most of my concerns.

  The authors indicated they made “Figure A. PCA of the top 2099 CpGs from the differential analysis between grade 1 and grade 2+3 meningiomas.”. But I couldn’t find it.

Author Response

Thank you for your comments. We have now added a new supplementary Figure S1. 

Reviewer 2 Report

Reviewer comments were adequately responded to.

Author Response

Thank you very much for your comments. Best regards